# Learning unfolded networks with a cyclic group structure

**Emmanouil Theodosis**                                      ETHEODOSIS@G.HARVARD.EDU
*School of Engineering and Applied Sciences*
*Harvard University*
*Cambridge, MA 02138*

**Demba Ba**                                                DEMBA@SEAS.HARVARD.EDU
*School of Engineering and Applied Sciences*
*Harvard University*
*Cambridge, MA 02138*

**Editors:** Sophia Sanborn, Christian Shewmake, Simone Azeglio, Arianna Di Bernardo, Nina Miolane

## Abstract

Deep neural networks lack straightforward ways to incorporate domain knowledge and are notoriously considered black boxes. Prior works attempted to inject domain knowledge into architectures *implicitly* through data augmentation. Building on recent advances on equivariant neural networks, we propose networks that *explicitly* encode domain knowledge, specifically equivariance with respect to rotations. By using unfolded architectures, a rich framework that originated from sparse coding and has theoretical guarantees, we present interpretable networks with sparse activations. The equivariant unfolded networks compete favorably with baselines, with only a fraction of their parameters, as showcased on (rotated) MNIST and CIFAR-10.

**Keywords:** Equivariance, model-based learning, cyclic groups, unfolded networks

## 1. Introduction

While advances in deep neural networks have yielded groundbreaking results in various fields such as computer vision (Redmon and Farhadi, 2017; Pavlakos et al., 2017; Mildenhall et al., 2020), natural language processing (Devlin et al., 2019; Brown et al., 2020), and their intersection (Radford et al., 2021), interpreting their structure and explaining their performance is not straightforward. At the same time, applying deep learning techniques to novel fields comes with challenges, as it is not clear how to integrate domain knowledge into existing architectures. In this work, we propose a novel architecture to address both of these shortcomings at the same time, leading to an interpretable architecture that respects expert knowledge.

Convolutional Neural Networks (CNNs) are *equivariant* in their representations with respect to translation; however, there are other operators that is natural for image models to be equivariant to, such as rotations. While data augmentation techniques have been used to model equivariances they require large amounts of data and increase the computational demands for training, frequently by an order of magnitude. At the same time, if we know the desired equivariances for a specific application, investing computational resources to re-learn these equivariances is wasteful. This was also acknowledged by Dieleman et al. (2016) and Cohen and Welling (2016) who concurrently introduced CNN frameworks that incorporate rotated filters in order to create equivariant representations with respect to

rotations; however both works were limited to elementary rotations and the reflections of $\mathcal{D}_4$. Follow up works by different authors extended the ideas to vector fields (Marcos et al., 2017), applied rotations directly on the sphere to avoid interpolation artifacts (Esteves et al., 2020), and incorporated harmonic functions to model equivariance with respect to arbitrary rotations (Worrall et al., 2017).

There have been several attempts to tackle interpretability, ranging from prototype learning approaches (Chen et al., 2019; Arik and Pfister, 2020) that learn *prototypical* parts for each class in a classification task, to post-hoc methods (Ribeiro et al., 2016) that reverse engineer predictions from arbitrary classifiers. In this work, we focus on *model-based* networks (Shlezinger et al., 2020): in these approaches, interpretability is directly encoded into the model by constructing a neural network to mimic the steps of an optimization algorithm. First introduced by Gregor and LeCun (2010), *unrolled* neural networks have inspired a vast array of works, ranging from theoretical contributions (Nguyen et al., 2019; Arora et al., 2015) to state-of-the-art results (Tolooshams et al., 2020).

In this work, we propose an unrolled architecture, inspired from algorithms for sparse coding, whose layer weights employ a *cyclic group structure* to achieve rotational equivariance. This allows us to significantly reduce the number of trainable filters compared to baseline architectures. Concretely, our contributions can be summarized as follows:

1. We propose an *unrolled* architecture, modeled after sparse coding, that is by construction interpretable and equivariant with respect to rotations,
2. we showcase its efficacy in learning filters that are governed by a cyclic group structure with a fraction of the learnable parameters, and
3. we evaluate the proposed architecture on MNIST, rotated MNIST, and CIFAR-10, standard benchmarks for rotationally equivariant architectures and demonstrate its performance.

We make the code for our experiments and architectures available online on GitHub at https://github.com/manosth/cyclical_groups.

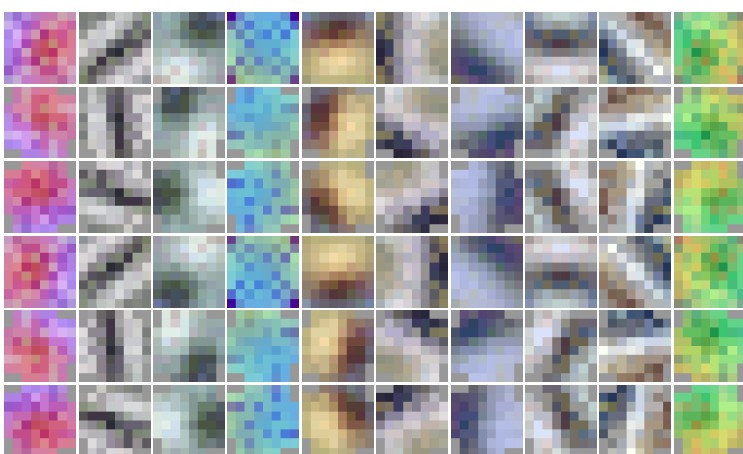

Figure 1: Filters learned at the final layer of $R_{60}$-CNN, training on CIFAR-10.

## 2. Background

**Equivariance.** In lay terms, an operator is equivariant with respect to some actions if it behaves in a predictable manner under them. Formally, we say that an operator $f$ is equivariant with respect to a family of actions $\mathcal{T}$ if, for any $T \in \mathcal{T}$ it holds that

$$f(T(x)) = T'(f(x)), \tag{1}$$

for some *other* transform $T'$. Note that, in general, $T$ and $T'$ belong in different families as $T$ acts on the input space $(\mathrm{Dom}(f))$ and $T'$ on the encoding space $(\mathrm{Im}(f))$. Constant functions are trivially equivariant, and a special case, *invariance*, arises when $T'$ is the identity map. Note that convolution is *not* equivariant to rotation (Cohen and Welling, 2016; Dieleman et al., 2016); instead, the two are related by

$$R(\boldsymbol{x}) * \boldsymbol{h} = R(\boldsymbol{x} * (R^{-1}(\boldsymbol{h})),$$

where $\boldsymbol{x}$ denotes an input image, $R$ is a rotation, and $\boldsymbol{h}$ is the convolving filter.

**Cyclic groups.** We call a finite group $G$ a cyclic group if there exists a generating element $g$ such that

$$G = \{e, g, g^2, \ldots, g^{k-1}\}, \tag{2}$$

where $e$ denotes the identity element. We denote the family of cyclic groups as $\mathcal{G}$; several groups belong to this family, with most notable being $\mathcal{D}_4$, the symmetry group of the square. Cyclic groups are of interest for our model since all elements can be identified by the generator $g$. This will enable us, in Section 3 to significantly reduce the trainable parameters of our networks, while retaining (and even improving) performance and interpretability.

**Unrolled sparse autoencoders.** In their most general form, unrolled networks temporally unroll the steps of optimization algorithms, mapping algorithm iterations to network layers. In that way, the output of the neural network can be interpreted as the output of the optimization algorithm, with theoretical guarantees under certain assumptions. The *Iterative Soft Thresholding Algorithn* (ISTA), an algorithm for sparse coding, has inspired several architectures (Simon and Elad, 2019; Sulam et al., 2020; Tolooshams et al., 2021), due to the desirability of sparse representations. Within that framework, the representation at layer $l + 1$ is given by

$$\boldsymbol{z}^{(l+1)} = \mathcal{S}_\lambda \left( \boldsymbol{z}^{(l)} + \alpha \boldsymbol{W}_l^T (\boldsymbol{x} - \boldsymbol{W}_l \boldsymbol{z}^{(l)}) \right), \tag{3}$$

where $\boldsymbol{x}$ is the *original* input, $\boldsymbol{z}^{(l)}$ is the representation at the previous layer, $\boldsymbol{W}_l$ are the weights of layer $l$, $\alpha$ is a constant such that $\frac{1}{\alpha} \geq \sigma_{\max}(\boldsymbol{W}_l^T \boldsymbol{W}_l)$, and $\mathcal{S}_\lambda$ is the *soft thresholding* operator defined as

$$\mathcal{S}_\lambda(u) = \mathrm{sign}(u) \cdot \mathrm{ReLU}(|u| - \lambda). \tag{4}$$

If the weights of all the $L$ layers are equal, i.e. $W_1 = \ldots = W_L$, we call the network *tied*. As a final remark, Equation (3) can be rewriten as

$$\boldsymbol{z}^{(l+1)} = \mathcal{S}_\lambda \left( (I - \alpha \boldsymbol{W}_l^T \boldsymbol{W}_l) \boldsymbol{z}^{(l)} + \alpha \boldsymbol{W}_l^T \boldsymbol{x} \right) = \mathcal{S}_\lambda \left( \boldsymbol{W}_z \boldsymbol{z}^{(l)} + \boldsymbol{W}_x \boldsymbol{x} \right), \tag{5}$$

where we let $\boldsymbol{W}_z = (I - \alpha \boldsymbol{W}_l^T \boldsymbol{W}_l)$ and $\boldsymbol{W}_x = \alpha \boldsymbol{W}_l^T$, which can be interpreted as a residual network (He et al., 2016), with a residual connection to the input.

## 3. Equivariant autoencoders

We will combine the ideas of cyclic groups and unrolled autoencoders from Section 2 to create an *equivariant* unrolled architecture, where the bulk of the weights in each layer are cyclic rotations of some "basis" weights . Let $R_\theta$ denote a two-dimensional rotation by $\theta$ degrees. If $360 \mod \theta = 0$ and we let $k = 360 \div \theta$, then the group defined as

$$G = \{e, R_\theta, \dots, R_\theta^{k-1}\}, \tag{6}$$

is a cyclic group generated by the generator $g = R_\theta$. This construction allows us to extend this framework, in future work, to *arbitrary* linear operators and the possibility to *learn* the generator $g$, leading to data-driven approaches for the cyclic group structure. Regardless, the weights of layer $l$ satisfy

$$\boldsymbol{W}_l = \begin{bmatrix} \boldsymbol{w}_l & R_\theta(\boldsymbol{w}_l) & \dots & R_\theta^{k-1}(\boldsymbol{w}_l), \end{bmatrix} \tag{7}$$

where $k$ is the number of rotations we are considering, per layer. Note that Equation (7) shows a single basis weight per layer; this can be readily extended to having $m$ basis weights $\boldsymbol{w}_{l_1}, \dots, \boldsymbol{w}_{l_m}$ along with the $k$ rotations for each of them, as is done in Section 4. The networks are constructed analogously to Sulam et al. (2020); Tolooshams et al. (2021) but with *one* (or $m$) learnable filter(s) per layer. During training, the remaining filters are constructed and errors are backpropagated through *all* filters. The experiments of Section 4 use untied networks for improved performance.

## 4. Experiments

We present experiments on MNIST, rotated MNIST, and CIFAR-10, standard benchmarks for equivariant architectures, to evaluate the architecture introduced in Section 3. We used batch normalization (Ioffe and Szegedy, 2015) in all of our architectures. The normalization was applied at the output of every layer, except the last. FISTA (Beck and Teboulle, 2009), a faster version of ISTA, is used for faster convergence of the sparse coding. All of our networks use $L = 4$ layers, $\lambda$ (the parameter of $\mathcal{S}_\lambda$) is set to 0.5, and the stepsize of FISTA in (3) is set to $\alpha = 0.01$. A summary of our main results is given in Table 1.

We test three models: an unrolled sparse network as a baseline; $R_{90}$-CNN, an equivariant unfolded network with the elementary rotations; and $R_{60}$-CNN, with 60° rotations. Every architecture has 60 filters per layer; in the case of the baseline, all 60 are learnable, $R_{90}$-CNN has 15 learnable filters per layer the rest are constructed as elementary rotations,

| Method | MNIST | rot-MNIST | CIFAR-10 | Number of parameters |
|---|---|---|---|---|
| Baseline | **99.21** | 85.48 | 71.87 | 56.29K |
| $R_{90}$-CNN | 99.12 | **86.62** | 72.20 | 21.73K |
| $R_{60}$-CNN | 98.77 | 80.07 | **73.14** | **17.89K** |

Table 1: Performances of the baseline model, $R_{90}$-CNN, and $R_{60}$-CNN on different datasets along with their total number of trainable parameters (CIFAR-10).

and $R_{60}$-CNN has only 10 learnable filters per layer with the rest being their 60° rotations. The learned filters are $7 \times 7$ in the case of MNIST and $8 \times 8$ in the case of CIFAR-10. A visualization of $R_{60}$-CNN's learned filters when trained on CIFAR-10 is show in Figure 1.

**MNIST.** We first evaluate all three models on MNIST, a relatively easy dataset, and find that all of them performed similarly. However, we note that $R_{90}$-CNN has only $\frac{1}{4}$ the trainable filters of the baseline, as the rest are generated as rotations of the basis weights; $R_{60}$-CNNH has only $\frac{1}{6}$. When evaluating the architectures on the rotated MNIST, a harder dataset, we observe that the $R_{90}$-CNN, with a *fraction* of the parameters of the baseline model leads to the best performance. This showcases that the encoded equivariance in the representation is actually beneficial for the classification of the inputs.

Evaluating the architectures on the rotated MNIST produces similar results and $R_{90}$-CNN outperforms the baseline. However, $R_{90}$-CNN displays reduced performance on this dataset; we speculate that this is because of interpolation artifacts, which was one of the motivating reasons for Esteves et al. (2020) to consider rotations on the sphere.

To further demonstrate the benefit of the equivariant unfolded networks, we trained *dense* variants (where the filters are now $28 \times 28$ images) of the three models on MNIST, and evaluate their performance on the rotated dataset. The results are presented in Table 2. This experiment showcases the generalization capabilities of the equivariant unrolled networks. While we see similar performance across all models on the trained dataset, we see that *both* the equivariant models are able to generalize better than the baseline. Dense architectures were chosen for this experiment to highlight the distribution shift when evaluating on rot-MNIST.

| Method | MNIST | rot-MNIST |
|--------|-------|-----------|
| Baseline | 97.75 | 36.89 |
| $R_{90}$-NN | **98.04** | 37.02 |
| $R_{60}$-NN | 97.73 | **37.4** |

Table 2: Models trained on MNIST but evaluated on rot-MNIST.

**CIFAR-10.** When training on CIFAR-10, a comparetively harder dataset, we found that *both* the equivariant models outperform the baseline, even if marginally, with only a fraction of the learnable filters per layer. Moreover, upon further examination, the filters of $R_{90}$-CNN and $R_{60}$-CNN exhibit a topographic structure by construction as filters are generated as rotations of one another. That topographic structure is not present in the filters of the baseline model which doesn't incorporate domain knowledge regarding rotations. The learned filters of the different models, when trained on CIFAR-10, are presented in Appendix A.

## 5. Conclusions and future work

In this work we introduced equivariant unrolled networks, where the filters of each layer are constructed as discrete rotations of one another. By exploiting this cyclical group structure, we are able to facilitate training and maintain performance while drastically reducing the number of trainable parameters of the model. We experimentally validated the equivariant unrolled networks against a baseline network and showed comparable, and even favorable, results, with only a fraction of the learnable parameters on multiple datasets that are used as benchmarks for equivariant architectures. Finally, we identify *learning* the generator $g$ from data, as hinted in Section 2, an exciting avenue for future work.

## Acknowledgements

Demba Ba was supported by the National Science Foundation under Grant number #S5206, PO 560825.

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

## Appendix A. Learned filters on CIFAR-10

We present filters learned on CIFAR-10 (without whitening) by the three architectures. We choose to present filters from the first layer, as those resemble edge detectors the most and thus are more straightforward to reason about. While all models seem to be learning

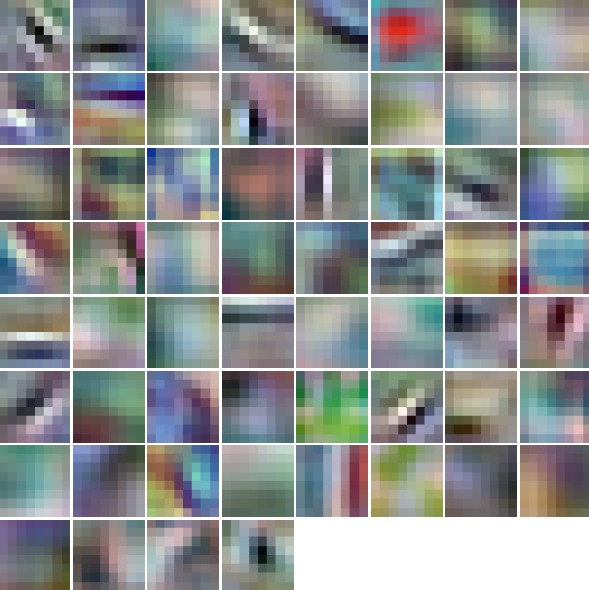

Figure 2: Filters learned using the baseline architecture.

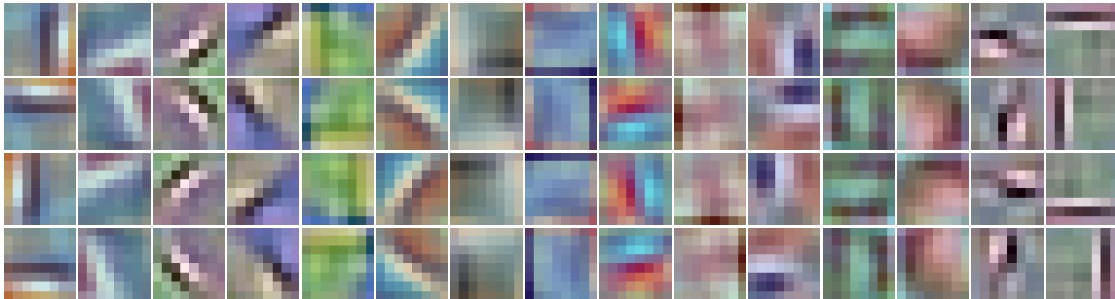

Figure 3: Filters learned using $R_{90}$-CNN.

similarly-looking filters, the equivariant models do not need to "waste" computation on learning different orientations. Indeed, if we observe Figure 2, the first filter from the top and the third from the bottom of the first column seem to be rotated versions of one another. In stark contrast, the third column of Figure 3 seems to learning the elementary rotations of that same filter, without investing resources on learning that information from the data. That is also observed in the first column of Figure 4, which has even more rotated versions of that same filter.

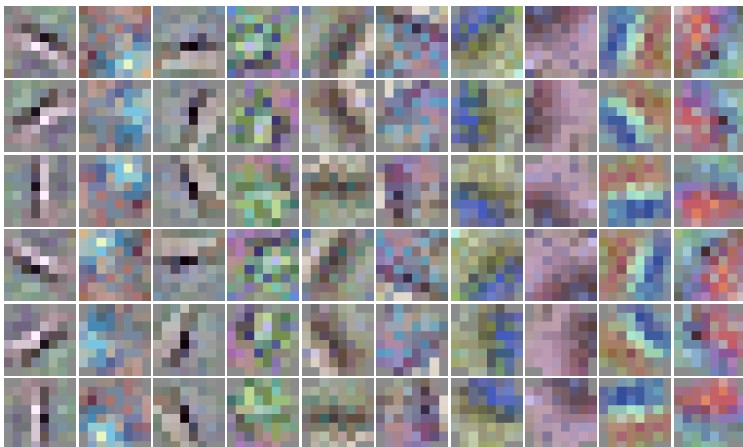

Figure 4: Filters learned using $R_{60}$-CNN.

