# OpenReview forum: "Learning unfolded networks with a cyclic group structure"
_NeurIPS.cc/2022/Workshop/NeurReps — NeurReps 2022 Poster_

### Official Review · Reviewer_LdVK · 2022-10-07
**Method not well motivated, nor evaluated**

**Confidence:** 4
**Soundness:** 2
**Presentation:** 3
**Contribution:** 2
**Overall Rating:** 5

**Summary:**

The authors design a network that is equivariant to rotation by a finite cyclic group. The architecture is inspired by Iterative Soft Thresholding, an algorithm used for sparse coding. A notable property is that the input to the network is fed to each layer in the network.

**Questions:**

Could you include the definition of the soft thresholding operator in the paper?

**Limitations:**

The statement in the abstract that the "networks compete favorably with baselines ... on MNIST and CIFAR-10" is perhaps a bit strong, as much better baselines exist.

**Recommended Decision:**

2: Borderline

**Relevance:**

3: Solid fit

**Strengths And Weaknesses:**

Strength:
- The proposed method is simple and inspired by useful algorithms.

Weakness:
- The authors don't provide sufficient explanation why the similarities between the neural architecture and the IST algorithm should lead to good performance when used for image classification.
- The proposed architecture uses a subset of all equivariant linear maps between the hidden representations as found in e.g. Cohen & Welling (2016). Concretely, the latent $z^l$ is always first mapped to the input representation, then mapped back to the lifted representation. This is not the most general map. Can the authors motivate why it is beneficial to not consider all possible maps?
- The experimental evaluation should include a comparison to other architectures.
- The final performance isn't very good. E.g. [1] finds >99% accuracy on rot-MNIST with cyclic equivariant methods and >96% on CIFAR-10 with rotation-mirror equivariant methods.

[1] Weiler, Maurice, and Gabriele Cesa. 2019. “General E(2)-Equivariant Steerable CNNs.”

**Submission Track:**

Extended Abstract (4 Page)

---

### Official Review · Reviewer_JWFn · 2022-10-12
**Motivation and references of this work are missing**

**Confidence:** 5
**Soundness:** 3
**Presentation:** 2
**Contribution:** 1
**Overall Rating:** 5

**Summary:**

This work presents unfolded architectures for sparse coding which exploit a rotation prior. Basically, covariant filters to rotation a learned by quotienting the filterbank and results are demonstrated on benchmarks for which rotation is an important variability.

**Questions:**

Can the authors explain which theoretical insights they obtain? How does their work compare with concurrent approaches?

**Limitations:**

Would the approach of the authors work for more complex groups like SO(3)? Probably not, given that there is no infinitesimal generator whose orbits is dense in the group. This work seems limited to groups spanned by a few infinitesimal generators. That could be discussed.

**Recommended Decision:**

1: Reject

**Relevance:**

2: Limited relevance

**Strengths And Weaknesses:**

I think this work lacks of motivation, originality and comparison with state-of-the-art approaches. Some possible theorerical improvements are mentioned, yet I'm unclear which novel perspective and insights are obtained thanks to this formulation, beyond a limited reduction in the model complexuty and minor improvements on dataset where rotation is an important variability. This type of approach is very similar to  [ https://hal.inria.fr/hal-02976813/file/deep_network_classification_by_scattering_and_homotopy_dictionary_learning_iclr_2020.pdf ] (and there are numerous examples of such works), which should be discussed as they follow a similar point of view. I agree that they also claim theoretical interpretability yet this is not fully justified. Consequently, I recommend the rejection of this work.

**Submission Track:**

Extended Abstract (4 Page)

---

### Official Review · Reviewer_nSxc · 2022-10-12
**Interesting new model that achieves comparable results to state-of-the-art with many less parameters. Authors need to re-focus results towards parameter size, as results *do not* indicate outperformance in classification..**

**Confidence:** 4
**Soundness:** 2
**Presentation:** 4
**Contribution:** 3
**Overall Rating:** 6

**Summary:**

In this work, authors share a new method for rotationally-equivariant unfolded networks. Their method ensures that each layer of a typical unfolded architecture is in fact the cyclic rotation of a previous layer, i.e. the architecture is made up of a cyclic group of layers with one generating layer.
They train two such models (with 90 and 60 degree rotations, respectively) and compare to a typical model. With 4 and 6 times less parameters respectively, the paper's models classify MNIST and CIFAR datasets with accuracy accuracy comparable to that of the typical model.

**Questions:**

- What dataset was used to train R90 and R60 in Table 1?
- Does the variable $L$ refer to the amount of layers in the network? The sentence "If W1 = ... = WL, we call the network tied." implies this is the case, but it not explicitly written anywhere.

**Limitations:**

- Results of this paper are limited in the sense that they only feature single experiments, so no statement can be made about definite outperformance of state-of-the-art in classification accuracy.
- The method is limited to 2D image data, as the implemented cyclic group structure only allows for 1 angle parametrization.

**Recommended Decision:**

2: Borderline

**Relevance:**

4: Highly relevant

**Strengths And Weaknesses:**

Originality: How novel and/or creative is the work?
- This is an innovative new architecture that develops on unfolded autoencoders in a mathematically informed way.
- While the performance of the model is not clearly better than its non-rotationally equivariant counterpart for classification, this work shows how equivariance can significantly reduce network size and could lead to better performances in classification.

Quality: Is the submission technically sound? Are claims well supported?
- This is my strongest issue with this paper. While the R90 and R60 models do show "better" classification than the baseline model in many cases, these are all tiny differences without uncertainties coming from multiple experiments (which could potentially show that these performances are in fact consistent and not different). Authors $\textbf{cannot claim that their models outperform baseline}$ with such results, and should rephrase the Experiments section to state that model performances are comparable (remove use of the word "outperform" in the classification sense). Considering the difference in amount of parameters, comparable performances are still a success and authors should make this the focus of the paper. (See comments in Clarity about parameter size)

Clarity: Is the submission well-organized and clearly written?
- The paper has a well written introduction and review of the math.
- Figures in the appendix are informative.
- In order to better drive home the point of drastic difference in parameter size, Table 1 should explicitly feature number of parameters (even if mentioned in the text). Authors could add a column for example with corresponding parameter amount or specify parameter amount in the caption.
- In section 3, the authors would benefit from explicitly writing that eq(6) means that layers are rotations of one another.

Significance: Are the results important and of interest to this community?
- This new architecture is an interesting and relevant result for geometric deep learning. However, it is imperative that authors $\textbf{do not overstate the classification performance}$ of their model relative to the typical unfolded autoencoder, as results are simply not indicative of outperformance (would need bigger differences and/or uncertainties from multiple experiments).

**Submission Track:**

Extended Abstract (4 Page)

---

### Decision · Program_Chairs · 2022-10-21

**Decision:**

Accept (Poster)

**Comment:**

The following extended abstract has received an average score of 5.33 (6,5,5) -- above the default cutoff for acceptance of extended abstracts -- but with a borderline-to-reject (2,1,2) recommended decision. Thus we review the assessment.

The work under consideration introduces a new architecture that exploits the concept of rotation-equivariance, by employing Iterative Soft Thresholding, to reduce the network size and to improve classification performances.

The main criticism of the paper concerns the erroneous comparison with the classification performance of previous models, and in particular that the model outlined here is not state-of-the-art - as noted by reviewer LdVK. Similarly, being an early stage work and not featuring a diverse set of experiments - as reported by reviewer nSxc - the model is far from being tested in depth on different tasks. Although these are fair criticisms for the authors to consider - we suggest making a fair comparison with SOTA models - we believe it is a secondary aspect in the evaluation of the extended abstract.

The work is well argued from a mathematical point of view and conceptually correct, with clear relevance for the computational neuroscience community. Even though it remains unclear how this architecture can be generalised to include more complex transformations and datasets - as highlighted by reviewer JWFn - we believe that bridging equivariant architectures and sparse coding is totally in line with the workshop topic and aim and of major interest for the community at the intersection of neuroscience and deep learning.

Moreover, we find the demonstrations on MNIST, rot-MNIST and CIFAR-10 sufficiently convincing for an early stage work to warrant inclusion as an extended abstract in the workshop. We recommend that the authors address critiques regarding SOTA results and referencing the line of research outlined by reviewer JWFn.